# A Cross-Sectional Study to Understand HPV Vaccine Hesitancy and Influencing Factors in Italian Adults

**DOI:** 10.3390/vaccines13060599

**Published:** 2025-05-31

**Authors:** Marianna Riccio, Azzurra Massimi, Erika Renzi, Michele Innocenzio, Roberta Siliquini, Fabrizio Bert, Anna Odone, Carolina Marzuillo, Paolo Villari, Corrado De Vito

**Affiliations:** 1Department of Public Health and Infectious Diseases, Sapienza University of Rome, 00185 Rome, Italy; azzurra.massimi@uniroma1.it (A.M.); erika.renzi@uniroma1.it (E.R.); michele.innocenzio@uniroma1.it (M.I.); carolina.marzuillo@uniroma1.it (C.M.); paolo.villari@uniroma1.it (P.V.); corrado.devito@uniroma1.it (C.D.V.); 2Department of Public Health and Paediatric Sciences, University of Turin, 10126 Turin, Italy; roberta.siliquini@unito.it (R.S.); fabrizio.bert@unito.it (F.B.); 3Department of Public Health, Experimental and Forensic Medicine, University of Pavia, 27100 Pavia, Italy; anna.odone@unipv.it

**Keywords:** human papillomavirus, vaccine hesitancy, Italy, survey

## Abstract

Objectives: The Human papillomavirus (HPV) vaccine is a cornerstone of cancer prevention, yet uptake remains suboptimal in many countries. This study analyzed the factors influencing HPV vaccine acceptance among adults, including a focused analysis of parental behaviors. Methods: Data were collected through a web-based survey using a questionnaire. We performed univariable analysis and three logistic regression analyses to investigate the determinants in the overall sample and among parents. Results: A total of 1821 participants were surveyed. HPV vaccination uptake was low, with only 6.9% of the total sample and 7.6% of young adults (18–35 yo) vaccinated. Among parents, 47.9% had vaccinated children aged 12–17, and 21.1% those aged 18 and over. Higher health literacy was associated with positive attitudes (OR 2.03, 95% CI 1.48–2.79), while receiving information from pediatricians or gynecologists was linked to children’s vaccination status (OR 7.30, 95% CI 2.29–23.31) and parents’ intentions for future HPV vaccination (OR 5.86, 95% CI 1.85–18.50). Adequate knowledge emerged as a strong predictor of positive attitudes (OR 6.50, 95% CI 4.91–8.61) and parents’ intentions (OR 4.89, 95% CI 2.21–10.84). Vaccination status was a key factor influencing parental decisions and overall vaccine acceptance. Conclusions: These findings highlight the critical role of health professionals and the need for targeted communication to address persistent knowledge gaps and promote HPV vaccine confidence within general population.

## 1. Introduction

The Human papillomavirus (HPV) is one of the most prevalent sexually transmitted infections globally. It poses a significant public health concern due to its established association with various cancers [1]. HPV vaccination has demonstrated a substantial impact in reducing the incidence of HPV-related malignancies and the prevalence of high-risk HPV types [2,3]. Despite its inclusion in national immunization programs, vaccination uptake remains suboptimal in several regions [4], potentially due to the knowledge, attitudes, and behaviors of the target population and their caregivers [5,6]. Successful immunization initiatives, such as those implemented in Australia and the United Kingdom, have shown the potential of widespread vaccination to reduce HPV-related morbidity markedly [7,8,9,10].

Italy launched its HPV vaccination program in 2007 for 12-year-old girls. It expanded in 2017 to include boys starting with the 2006 birth cohort, aiming to reduce transmission and disease burden among both sexes [11]. The vaccine is free of charge to girls and boys aged 11, as well as specific high-risk groups, and there are catch-up programs for females up to age 26 and males up to age 18 [12]. Nonetheless, vaccination coverage remains below both national and international targets. Recent surveillance data from 2023 reports coverage rates of only 45.4% among 12-year-old girls and 39.4% among boys of the same age, significantly below the 95% target set by the Italian National Immunization Plan [13]. These statistics highlight a critical gap in HPV vaccine uptake in Italy and emphasize the urgent need to evaluate awareness of HPV infection and vaccination among both parents and the general population to support comprehensive prevention efforts effectively.

Addressing low vaccine uptake requires a thorough analysis of the factors underlying vaccine hesitancy (VH) [14], a phenomenon shaped by individual beliefs, social norms, and structural barriers [15]. Vaccine hesitancy, whether general or vaccine-specific, threatens public health efforts, including the global initiative to eliminate cervical cancer [16]. Numerous studies have explored HPV-vaccine hesitancy in adolescents, parents, young adults, and healthcare professionals, identifying determinants such as gender, education level, sexual behaviors, and informational needs [5,6,17,18,19,20,21,22,23,24,25]. Building on this literature, the present study assesses the knowledge, attitudes, and behaviors of the Italian adult population regarding HPV infection and vaccination, to identify determinants to inform targeted, evidence-based public health strategies [12].

## 2. Materials and Methods

### 2.1. Study Design and Participants

The study used a cross-sectional design following the Strengthening the Reporting of Observational Studies in Epidemiology (STROBE) model [26]. Data were collected between February and March 2024 through a web survey. A random sample of Italian adults (≥18 years) was recruited through a stratified sampling strategy based on age, gender, and geographical area, using Norstat’s web panel. The questionnaire was based on the scientific literature and designed to take approximately five to six minutes. A pilot test was conducted for clarity and usability.

### 2.2. Survey Instrument

The first section of the questionnaire focused on socio-demographic data, including age, gender, nationality, geographical area, marital status, educational level, parenthood status, cohabitants with disabilities, occupation, presence of chronic diseases, monthly income, and financial resources. Political orientation, religious affiliation, and perception of the quality of the National Health System (NHS) were also assessed.

The second section included sources of information about HPV vaccination, self-efficacy (SE), and health literacy (HL). HL was measured using the validated Italian version of the Single-Item Literacy Screener (SILS) [27,28]. SE was assessed through three items rated on a 10-point Likert scale: perceived health self-management skills, perceived health status, and inclination to minimize healthcare utilization.

The third and fourth sections focused on HPV-specific knowledge, attitudes, and behaviors. Seven questions, rated on a 10-point Likert scale, investigated knowledge and attitudes regarding the perceived risk and severity of infection, the perceived safety of the vaccine, and its effectiveness and utility in cancer prevention. We collected data on vaccination status for participants and their children, including reasons for non-vaccination.

The internal consistency of the attitude and knowledge scales was evaluated using Cronbach’s alpha. The results showed a value of 0.82 for the knowledge scale (3 items) and 0.61 for the attitudes scale (4 items). Although the alpha coefficient for attitudes was slightly below the conventional 0.70 threshold, it was deemed acceptable given the small number of items and the complexity of the construct. Consistent with the prior literature [29], values above 0.60 are considered adequate in such contexts. Item analysis showed no benefit from item removal; thus, all were retained.

### 2.3. Statistical Analysis 

#### 2.3.1. Variables and Measures

Certain variables were recategorized for analysis. Financial status, based on the ability to manage household financial resources and assessed on a 5-point Likert scale, was dichotomized into medium–high and low. Political orientation was assessed initially on a 10-point scale and then categorized into four groups: left (1–3), moderate (4–7), right (8–10), and I prefer not to answer. Participants’ religious beliefs were classified as follows: Catholic, other religions (Orthodox, Jewish, Islamic, Jehovah’s Witness, others), none, and I prefer not to answer. The perception of NHS quality was categorized into three levels: poor (low quality), average (medium quality), and good (high quality). HL level was dichotomized as high and low, based on whether help was needed to understand medical information. Finally, based on their first preference, information sources were organized into four groups: pediatricians/gynecologists, HCWs besides pediatricians/gynecologists, other sources, and no information received. The category “other sources” included friends, family, acquaintances, mass media, and the internet.

Knowledge and attitudes, initially encompassing multiple categories assessed on a 0–10 scoring scale, were collapsed into two levels: those who agreed with all the three correct responses about HPV vaccination (showing good knowledge) versus all others; those who demonstrated positive attitudes in at least three of four items assessing attitudes (indicating positive attitudes) compared to all others.

Vaccination status was categorized into vaccinated and unvaccinated. Parental intention to vaccinate children aged 0–11 years was dichotomized using a 90-point threshold on a 0–100 scale.

To guide the interpretation of the results, we drew on the 5C model of psychological antecedents of vaccination behavior [30]. According to this model, the decision-making process that guides vaccination behavior is influenced by Confidence (trust in the efficacy and safety of vaccines, as well as in the health system), Complacency (perceived risk of vaccine-preventable diseases), Constraints (perceived or real barriers to vaccination), Calculation (deliberation and information seeking), and Collective Responsibility (willingness to protect others through vaccination).

#### 2.3.2. Statistical Methods

Descriptive statistics summarized categorical and continuous variables. For continuous variables, results were expressed as mean and standard deviation (SD) or median and interquartile range (IQR), depending on distribution. Categorical variables were reported as frequencies and percentages.

Each variable was analyzed using univariable analysis: Pearson’s chi-squared or Fisher’s exact test for dichotomous and categorical variables, and the Student’s *t*-test or Mann–Whitney U test for comparing continuous variables.

Three multivariable logistic regression models were developed based on the strategy suggested by Hosmer and Lemeshow [31] to identify predictors of (1) positive attitudes (general population), (2) children’s vaccination status (parents of children aged 12–17), and (3) intention to vaccinate children (parents of children under 12). Variables with a *p*-value ≤ 0.25 in univariable analysis were included in logistic regression models, alongside variables selected based on the literature and expert opinions. Goodness of fit was assessed using the Hosmer–Lemeshow test, and multicollinearity was checked using a variance inflation factor of 2.5 and a tolerance of 0.5. In Model 1, the gender category “I prefer not to answer” included only three observations. In Model 2, the gender category “I prefer not to answer” and the occupational category “students” comprised a single observation each. In Model 3, the occupational categories “students” and “retired” each included two observations. Due to the minimal sample sizes, these categories were excluded from regression analyses to avoid statistical and methodological issues.

#### 2.3.3. Study Size

The sample size was calculated based on the perception of HPV vaccine safety, a key determinant in vaccine hesitancy. Perceived vaccine safety is a well-established predictor of vaccine uptake and plays a central role in immunization decision-making. This approach aligns with the WHO’s Vaccine Hesitancy Determinants Matrix, which identifies safety concerns as a critical barrier to vaccine acceptance. Trust in vaccine safety also defines the “confidence” dimension of the widely recognized “3 Cs” model [14]. By prioritizing this variable, we ensured that our study could address HPV attitudes and identify the most critical and modifiable barriers to vaccine uptake and acceptance, thereby informing public health strategies aimed at improving vaccination coverage [12]. To calculate the appropriate sample size, we referenced data from a recent Italian study on young adults, which indicated that 32% of participants strongly agreed that the HPV vaccine is safe [32]. Assuming an expected proportion of 37%, an alpha error of 0.05, and a study power of 80%, we determined that a minimum of 718 participants would be required to detect statistically significant differences. The survey was initially designed to address multiple research objectives beyond the scope of the present analysis. Therefore, a larger sample size was targeted to ensure sufficient statistical power across different planned investigations and to allow for subgroup analyses. Ultimately, 1821 participants were included, providing a robust dataset for statistical analyses and allowing for more precise estimation of factors associated with VH in the Italian adult population.

All statistical calculations were performed using Stata version 18.0 software (StataCorp LLC, 4905 Lakeway Drive, College Station, TX, USA). A *p*-value < 0.05 was considered statistically significant.

## 3. Results

### 3.1. Description of the Study Population

The main characteristics of the sample are presented in Table 1 and Table 2. A total of 1821 participants completed the survey: 32.8% from the South/Islands, 27.5% from the Northwest, 20.4% from the Center, and 19.2% from the Northeast of Italy; 44% were female. Most participants were Italian (98.7%) and married (53.4%), with a mean age of 54 years (SD 17.2). Approximately one-third of the respondents had no children, while among parents, 46.1% had children aged 18 and over. In terms of education, around one-third had at least a university degree. Most participants were employed in non-healthcare sectors, and 29.2% were retired. Among HWCs (*n* = 55), 23.6% were physicians and 31% were nurses. Regarding financial status, 42% reported making ends meet fairly well, while 30.6% did so with some difficulty. Participants were also asked about their political orientation and religious affiliation, with about one-third identifying as politically moderate and 73.6% as Catholic. Most participants rated the quality of the NHS as medium. In the SE assessment, the highest-rated dimension was perceived health self-management skills. In terms of HL, it was reported that 30.5% and 37.6% of participants never or rarely needed assistance in understanding medical materials.

### 3.2. Knowledge, Attitudes, and Behaviors Towards the HPV Vaccine

Knowledge about the usefulness of the HPV vaccine showed a mean value of 7 (±2.4) for cervical cancer prevention and 6 (±2.6) for both oral cancer prevention and disease prevention in males (Table 3). Overall, 35% of participants demonstrated good knowledge about HPV vaccination (Appendix A). Positive attitudes were reported regarding both the safety and efficacy of the HPV vaccine, with a mean score of 7 (±2.3). A similarly positive, though slightly lower, score was observed for perceived risk of HPV infection (mean value 6 ± 2.7) (Table 3). The overall attitude level, calculated from five items, indicated that 19% of respondents had an overall positive attitude (Appendix A). The most frequently cited primary source of information was HCWs (30.8%)—which included general practitioners, NHS vaccination outpatient clinics, and others—followed by the mass media/internet (17.3%), and pediatricians/gynecologists (10.3%). Notably, over one-third of the sample (37.3%) reported never having received information about or heard of HPV vaccination (Table 2). HPV vaccination coverage in our study population was 6.9%, with the majority having received the full schedule. Among participants aged 18–35 years, coverage was 7.6% (Table 4). Among those unvaccinated at the time of the survey, the most frequently cited reasons included the unavailability of the vaccine and a lack of awareness about the opportunity, either on their part or that of their parents (Table 4). Among parents with children aged 12–17, 47.9% reported vaccinating them, compared to 21.1% of those with children aged 18 and over (Table 4). Among parents who had not vaccinated their children (*n* = 760), the most cited reasons were a lack of awareness about the vaccination and having only male children (Appendix A). The mean probability of future vaccination for parents with children aged 0–11 years (*n* = 269) was 52% (±38.2) (Table 4).

### 3.3. Univariable Analysis

Based on the univariable analyses (Appendix A), no socio-demographic characteristics—including gender, nationality, age, marital status, number of children, geographical area, or educational level—were significantly associated with positive attitudes toward HPV. Similarly, no significant differences were observed in personal monthly income or perceived financial resources between participants with and without positive attitudes. However, occupation was significantly associated (*p* = 0.002), with higher proportions of HCWs, retired individuals, and students among those reporting positive attitudes. Significant differences also emerged in two of the three SE items. These referred, respectively, to the individual’s confidence in effectively managing their health and to a tendency to deal with health problems independently, without frequently turning to healthcare services. The HL was found to be significantly higher (*p* < 0.001) among those with positive attitudes (81%) compared with those without (65%). Similarly, participants with a positive attitude reported a good perception of the NHS quality (49% vs. 37%). Those with a positive attitude were more inclined to have good knowledge than those without (73% vs. 26%, *p* < 0.001). Notably, sources of information showed significant variation by attitude level (*p* < 0.001): a greater proportion of participants with positive attitudes reported receiving information from pediatricians, gynecologists, or other healthcare providers, whereas those with negative attitudes more frequently reported having received no information at all. HPV vaccination coverage—both personal and in children (aged 12–17, and aged 18 and over)—was higher among subjects with positive attitudes (*p* < 0.001). Furthermore, a statistically significant association of positive attitudes was found with the intention to vaccinate children under 12 years of age in the future.

### 3.4. Multivariable Logistic Regression

#### 3.4.1. General Population (Model 1)

The model was developed using the entire survey sample (*n* = 1818), with the positive attitude level defined as the outcome of interest (Table 5). Results from the multivariable logistic regression analysis identified four independent variables significantly associated with this outcome. Positive attitudes were statistically significantly higher among individuals who reported receiving information from pediatricians or gynecologists (OR = 1.68; 95% CI 1.06–2.62, *p* = 0.026) as well as from other HCWs (OR = 1.56; 95% CI 1.11–2.19, *p* = 0.011), compared to those who had not received any information. Participants previously vaccinated against HPV, those with a high level of HL, and those with adequate knowledge had a more positive attitude.

#### 3.4.2. Parents (Model 2)

This model focused on parents with children aged 12 to 18 (*n =* 232), using children’s HPV vaccination status as the outcome of interest (Table 6). Results from the multivariable logistic regression model identified three independent variables significantly associated with this outcome. Parents who received information from their pediatrician/gynecologist vaccinated their children (OR = 7.30, 95% CI 2.29–23.31, *p* = 0.001) more than those without information. Parents who had themselves received the HPV vaccine were significantly more likely to have their children vaccinated (OR = 29.43, 95% CI: 3.34–259.41, *p* = 0.002). Finally, positive attitudes were associated with vaccinating their children (OR = 4.56; 95% CI: 1.65–12.60, *p* = 0.003).

#### 3.4.3. Parents (Model 3)

The third model examined the intention to vaccinate one’s child as the outcome of interest among the parents with children under 12 years of age (*n* = 265) (Table 7). Parents who received information from pediatricians/gynecologists (OR = 5.86; 95% CI 1.85–18.50, *p* = 0.003), as well as those who had received the HPV vaccine (OR = 4.07; 95% CI 1.48–11.19, *p* = 0.006), were more likely to be willing to vaccinate their child in the future. Left political orientation (OR = 0.22; 95% CI 0.07–0.86, *p* = 0.007) was negatively associated with the vaccination status of children compared to those with a moderate orientation. Additionally, a more favorable perception of NHS quality was associated with an increased willingness to vaccinate their children (OR = 4.09; 95% CI: 1.04–16.00, *p* = 0.043). Finally, both adequate knowledge (OR = 4.89; 95% CI 2.21–10.84, *p* < 0.001) and positive attitudes (OR = 4.90; 95% CI 2.02–11.84, *p* < 0.001) resulted as positive predictors.

## 4. Discussion

This large-scale study provides insights into HPV vaccine acceptance both within the general population and among parents, emphasizing the need for targeted, multifactorial interventions to raise awareness of HPV and vaccination uptake. Our findings contribute to the existing scientific literature by exploring knowledge, attitudes, and behaviors, mainly focusing on the specific factors influencing these dimensions. We align with the existing literature while identifying context-specific challenges that require specific public health interventions. Additionally, the subgroup analysis allowed for a comprehensive review of the specific characteristics influencing parents’ practices, underscoring their pivotal role in initiating and adhering to HPV vaccination schedules. As primary health decision-makers for their children, they significantly affect vaccine acceptance and uptake.

Not surprisingly, knowledge about the HPV vaccine’s role in cancer prevention was higher for cervical cancer than for other HPV-related diseases, such as those in males. This aligns with prior research emphasizing female-focused vaccination strategies [18,20,23,33] and the limited recognition of HPV’s association with conditions [18,19,33,34,35] such as oropharyngeal cancer and genital warts. Improving public understanding of the broader health implications of HPV is critical—not only to reduce VH but also to support vaccination programs aimed at eliminating all HPV-related diseases in high-income countries [12,36]. These findings contrast with the substantial burden of HPV infections affecting both sexes [37], reinforcing the need for campaigns that communicate the full scope of HPV-related morbidity and mortality [38]. Indeed, our findings regarding perceived risk and severity suggest that HPV vaccination campaigns should further address complacency, one of the recognized key aspects of VH [14]. While our findings are consistent with some studies [17,20,32,39], they diverge from others [33,40,41] in which the severity of HPV was more widely acknowledged. Risk perception is shaped by individual and contextual factors and should be interpreted accordingly [42]. These dimensions are well-established in the Vaccine Hesitancy Determinants Matrix developed by the SAGE Working Group [14] and should be considered by public health authorities.

Our findings highlight HL’s essential role in shaping attitudes towards HPV. Health literacy refers to people’s ability to obtain, process, and understand basic information and services to make informed health decisions [43]. Recent literature has explored its role as a health determinant, noting, for example, its association with lower utilization of health services and the development of specific health outcomes [44,45]. In recent years, the scientific community has increasingly explored HL in the context of HPV infection and vaccination, and several findings are consistent with studies conducted in Italy and internationally [23,46,47,48,49,50]. For instance, our results corroborate those of Kitur et al. [48], who identified a strong association between HL and vaccine confidence, highlighting the role of educational strategies to enhance understanding of vaccination benefits. Furthermore, the existing literature about HL and vaccine uptake, as reported by Falluca et al. [23] and Bhoopathi et al. [49], indicated that individuals with higher HL were significantly more likely to have received the HPV vaccine.

In our sample, HCWs were the most frequently cited primary source of information, consistent with numerous studies identifying their central role [18,20,25,32,51], particularly pediatricians among parent samples [18,33,52], as the primary source. Participants who received HPV information from pediatricians and/or gynecologists were significantly more likely to show positive attitudes, to have vaccinated children, and to vaccinate their children in the future. Our results reinforce this notion, especially those directly engaged with HPV-targeted populations. Previous studies have reported associations between these sources and positive perceptions about HPV and its vaccination [18,32,52,53]. Trucchi et al. [32], for example, found higher knowledge scores among individuals informed by obstetricians, and higher attitude scores among those citing pediatricians or general practitioners. A recent systematic review and meta-analysis also demonstrated the impact of provider communication on HPV vaccination initiation, completion, and follow-through [54]. Once again, these findings emphasize the importance of information campaigns led by adequately trained HCWs as a potentially effective strategy for achieving immunity goals and disseminating evidence-based information.

The internet and media also emerged as key information sources following HCWs and should not be underestimated, as they have been identified as primary sources in several previous studies [48,51,55,56]. Finally, nearly one-third of respondents reported receiving no information about HPV at all. This is concerning, as a lack of information is a known driver of vaccine hesitancy [57]. Public health authorities must address this gap, as awareness and understanding of HPV vaccination are closely tied to uptake [33,48,50,52]. Additionally, we observed that greater knowledge correlates with more favorable attitudes among the general population and with stronger parental intentions to vaccinate their children in the future.

Among parents, reported vaccination coverage for children was low, and uncertainty about future decisions was shared. The main reasons for not vaccinating included limited awareness and the belief that vaccination is unnecessary for boys—findings that are aligned with other studies [33,52,58]. A Centers for Disease Control and Prevention (CDC) report on unvaccinated adolescents similarly found that parental refusal was mainly due to insufficient recommendations, limited knowledge, and doubts about necessity [59]. To address these issues, school-based educational programs are crucial, as such initiatives can simultaneously increase awareness and acceptance among adolescents—the primary target group—and their parents, who are responsible for vaccination decisions [60]. Community Health Nurses (CHNs) could also be pivotal in providing direct care and supporting public health objectives through tailored educational initiatives [61]. These community interventions would improve access to vaccination services, a key component of the Immunization Agenda 2030 [62], and advance broader public health goals [16].

Finally, although potentially relevant, we found no significant overall results regarding religious and political orientation. For instance, a recent meta-analysis suggests that religious beliefs may influence HPV vaccination decisions [63]. In our study, political orientation significantly predicted intentions only among parents of children under 12, with liberal-leaning parents reporting lower intentions—a finding that contrasts with previous research [63,64] and may reflect the influence of unmeasured confounding variables. 

In conclusion, the findings of our study appear consistent with the theoretical construct of the 5C model [29]. The relevance of Confidence is evident, particularly in the central role played by trusted HWCs, such as pediatricians and gynecologists, in influencing vaccination behavior. The limited perception of risk—especially concerning male susceptibility—highlights the dimension of Complacency. Moreover, the associations between HL, knowledge, and vaccination intentions underscore the relevance of Calculation. Although not all model dimensions were directly assessed, the 5C framework provides a valuable lens through which to interpret the multifaceted nature of VH.

Several limitations should be considered. First, the cross-sectional design limits causal inference, and future longitudinal studies could explore how vaccine attitudes evolve and how interventions affect behaviors. Second, respondents may have provided socially acceptable answers, especially regarding vaccination status. Future research should incorporate qualitative methods like focus groups to gain deeper insights into decision-making processes. Third, there is a potential selection bias, as only individuals from the recruitment panel were included in the study. Additionally, while the large sample size increased the statistical power of our analyses, it may also heighten the chance of detecting statistically significant associations that lack substantial practical importance. Therefore, interpretation of results, especially those with borderline significance, should be cautiously approached. Finally, parents are underrepresented, since this population’s sample size was not statistically powered. Therefore, the findings related to parental vaccine decision-making may not fully reflect the perspectives of the broader parental population. We also acknowledge the limitation of using a validated single-item measure of HL, although more comprehensive tools may yield more profound insights. Despite these limitations, this study is based on a large sample and remains one of the few investigations exploring HPV-related attitudes in the general Italian population.

## 5. Conclusions

Our findings underscore the critical role of health literacy and competent healthcare professionals in influencing attitudes and behaviors related to HPV vaccination. We identified significant issues related to gaps in attitudes, knowledge, and practices and their underlying determinants, reinforcing the existing scientific literature. Future health programs should prioritize enhancing the dissemination of evidence-based information through trusted sources like pediatricians and gynecologists to improve HPV vaccine acceptance and uptake.

## Figures and Tables

**Table 1 vaccines-13-00599-t001:** Demographic and professional characteristics of the respondents.

***N* = 1821**	
	Mean (SD)
**Age**	54 (17.2)
**Age groups**	N (%)
18–24	83 (4.6)
25–34	256 (14.1)
35–44	266 (14.6)
45–54	277 (15.2)
55–64	279 (15.3)
≥65	660 (36.2)
**Gender**	
Male	832 (45.8)
Female	986 (54.2)
I prefer not to answer	3 (0.2)
**Nationality**	
Italian	1797 (98.7)
Non-Italian	24 (1.3)
**Geographical area**	
Northwest	501 (27.5)
Northeast	350 (19.2)
Center	372 (20.4)
South/Islands	598 (32.8)
**Marital status**	
Single	431 (23.7)
Married	972 (53.4)
Separated/Divorced	134 (7.4)
Cohabiting	209 (11.5)
Widowed	75 (4.1)
**Educational level**	
High school and low	1237 (67.9)
University degree	453 (24.9)
Postgraduate	131 (7.2)
**Having children**	
No	596 (32.7)
Yes, 0–11 yo	269 (14.8)
Yes, 1217 yo	234 (12.9)
Yes, ≥18 yo	839 (46.1)
**Cohabitants with disabilities**	
Yes	307 (16.9)
No	1514 (83.1)
**Chronic disease**	
No	880 (48.3)
Yes	941 (51.7)
**Occupation**	
Non-healthcare occupation	873 (48.0)
Healthcare workers	55 (3.0)
Housewives	195 (10.7)
Retired	531 (29.2)
Student	39 (2.1)
Unemployed	128 (7.0)
**Specific Health Occupation** (*n* = 55)	
Medical Doctor	13 (23.6)
Nurse	17 (31.0)
Healthcare Assistant	7 (12.7)
Other healthcare professions	18 (32.7)
**Which of these values is closest to your monthly personal income?**	
I have no income	193 (10.6)
Less than EUR 1000	276 (15.1)
EUR 1000–1499	455 (25.0)
EUR 1500–1999	454 (24.9)
EUR 2000–2999	321 (17.6)
EUR 3000–4499	87 (4.8)
EUR 4500–5999	23 (1.3)
Over EUR 6000	12 (0.7)
**With the financial resources available in your household, how do you make ends meet?**	
Definitely good	134 (7.4)
Fairly well	764 (42.0)
With some difficulty	721 (39.6)
With many difficulties	202 (11.1)
**Religion**	
Catholic	1340 (73.6)
Other religions	69 (3.8)
None	354 (19.4)
I prefer not to answer	58 (3.2)
**Importance of religion** (*n* = 1467)	
High	695 (47.4)
Medium	417 (28.4)
Low	316 (21.5)
I prefer not to answer	39 (2.7)
**Political orientation**	
Left	465 (25.5)
Moderate	586 (32.1)
Right	426 (23.4)
I prefer not to answer	344 (18.9)
**Perceived quality of the National Healthcare System**	
Good (high level)	708 (38.9)
Average (medium level)	840 (46.1)
Poor (low level)	273 (15.0)

**Table 2 vaccines-13-00599-t002:** Information sources, health literacy, and self-efficacy.

Self-Efficacy	Mean Score (SD)
Perceived health self-management skills	6.7 (1.6)
Perceived health status	6.2 (2.2)
Attempt to make little use of healthcare services when needed	5.3 (2.5)
**“How often do you need somebody to help you read instructions, flyers or other materials that you were given by your doctor or pharmacist?”**	N (%)
Never	556 (30.5)
Rarely	685 (37.6)
Sometimes	483 (26.5)
Often	85 (4.7)
Always	12 (0.7)
**Sources of information about HPV**	
Healthcare workers	560 (30.8)
Pediatrician/gynecologist	188 (10.3)
Friends, family, and/or acquaintance	67 (3.7)
Mass media/internet	315 (17.3)
Other	12 (0.7)
I have never received information about or heard of HPV vaccination	679 (37.3)

**Table 3 vaccines-13-00599-t003:** Knowledge and attitudes towards HPV vaccination (assessed on a 0–10 scoring scale).

QUESTIONS	Mean (SD)
“How much do you believe that HPV vaccines are safe?”	7 (2.3)
“How much do you believe that HPV vaccines are effective?”	7 (2.3)
“I believe that I am not particularly at risk of contracting HPV infection”	6 (2.7)
“I believe that HPV infection does not cause serious illness”	4 (2.6)
“HPV vaccination is useful in preventing cervical cancer”	7 (2.4)
“HPV vaccination is useful in preventing oral cancer (mouth)”	6 (2.6)
“HPV vaccination is useful in males”	6 (2.6)

**Table 4 vaccines-13-00599-t004:** HPV vaccination practices of the respondents.

N (%)	
**Have you ever received HPV vaccination?**	
Yes	125 (6.9)
No	1492 (81.9)
Don’t know/don’t remember	204 (11.2)
**If vaccinated, how many vaccine doses have you received?** (*n* = 125)	
Complete vaccine schedule	103 (82.4)
Incomplete vaccine schedule	6 (4.8)
Don’t know/don’t remember	16 (12.8)
**If you had to choose again whether you wanted to get vaccinated for HPV or not, what would you do?** (*n* = 125)	
Yes	105 (84.0)
No	11 (8.8)
Don’t know	9 (7.2)
**Have you ever received HPV vaccination?** (*aged* 18–35 *years*, *n* = 381)	
Yes	29 (7.6)
No	303 (79.5)
Don’t know/don’t remember	49 (12.9)
**If children (12–17 yo), did you vaccinate your child for HPV?** (*n* = 234)	
Yes	112 (47.9)
No	122 (52.1)
**If children (≥18 yo), did you vaccinate your child for HPV?** (*n* = 839)	
Yes	177 (21.1)
No	662 (78.9)
	Mean (SD)
**If children 0–11 yo, from 0 to 100 how likely is it that you will vaccinate your children for HPV?** (*n* = 269)	52 (38.2)
**If No/Not all of my children have been vaccinated, from 0 to 100 how likely is it that you will vaccinate your children for HPV in the future?** (*n* = 786)	26 (29.5)

**Table 5 vaccines-13-00599-t005:** Results of the binary logistic regression (Model 1). Factors associated with positive attitudes towards HPV vaccination *.

Variables	OR (95% CI)	*p*-Value
**Age**	1.00 (0.99–1.01)	0.915
**Gender**		
Male	Ref	
Female	0.90 (0.69–1.18)	0.441
**Educational level**		
High school or lower	Ref	
University degree	1.01 (0.74–1.38)	0.936
Postgraduate	0.98 (0.59–1.64)	0.951
**Ability to manage household financial resources**		
Definitely good/Fairly well	Ref	
With some difficulty/with many difficulties	1.07 (0.82–1.41)	0.604
**Occupation**		
Healthcare workers	Ref	
Non-healthcare occupation	0.56 (0.29–1.08)	0.084
Housewives	0.73 (0.35–1.52)	0.397
Retired	0.79 (0.41–1.54)	0.492
Students	1.16 (0.41–3.29)	0.779
Unemployed	0.72 (0.33–1.61)	0.427
**Source of information**		
No information received	Ref	
Other HCWs (other than pediatrician/gyn)	1.56 (1.11–2.19)	0.011
Pediatrician/gyn	1.67 (1.06–2.62)	0.026
Other sources	1.32 (0.91–1.92)	0.145
**Have you ever received HPV vaccination?**		
No	Ref	
Yes	1.66 (1.04–2.65)	0.034
**Self-Efficacy** (ability to take care of one’s own health)		
**No**	Ref	
**Yes**	1.07 (0.98–1.17)	0.142
**Health Literacy**		
Low	Ref	
High	2.03 (1.48–2.79)	<0.001
**Religion**		
None	Ref	
Catholic	1.19 (0.82–1.70)	0.358
Other	0.86 (0.41–1.79)	0.688
I prefer not to answer	1.14 (0.49–2.67)	0.766
**Political orientation**		
Moderate	Ref	
Left	1.16 (0.81–1.65)	0.421
Right	1.13 (0.80–1.61)	0.488
I prefer not to answer	0.77 (0.51–1.16)	0.210
**Perceived quality of the Healthcare System**		
Poor (low quality)	Ref	
Average (medium quality)	1.04 (0.68–1.60)	0.844
Good (high quality)	1.32 (0.86–2.02)	0.211
**Knowledge of HPV vaccination**		
Limited	Ref	
Good	6.50 (4.91–8.61)	<0.001

* outcome of interest: positive attitudes (343, 18.8%) vs. negative attitudes (1.378, 82.2%). Ref: reference category.

**Table 6 vaccines-13-00599-t006:** Results of the binary logistic regression (Model 2). Factors associated with HPV vaccination of children aged 12 to 17 *.

Variables	OR (95% CI)	*p*-Value
**Age**	1.01 (0.99–1.03)	0.221
**Gender**		
Male	Ref	
Female	0.65 (0.33–1.38)	0.209
**Educational level**		
High school or lower	Ref	
University degree	1.29 (0.54–3.06)	0.556
Postgraduate	2.57 (0.69–9.86)	0.170
**Ability to manage household financial resources**		
Definitely good/Fairly well	Ref	
With some difficulty/with many difficulties	1.01 (0.52–1.96)	0.979
**Occupation**		
Healthcare workers	Ref	
Non-healthcare occupation	2.72 (0.47–15.78)	0.265
Housewives	5.89 (0.72–48.46)	0.099
Retired	1.49 (0.13–19.39)	0.749
Unemployed	1.84 (0.20–16.95)	0.590
**Source of information**		
No information received	Ref	
Other HCWs (other than pediatrician/gyn)	1.25 (0.55–2.86)	0.595
Pediatrician/gyn	7.30 (2.29–23.31)	0.001
Other sources	1.66 (0.63–4.42)	0.309
**Have you ever received HPV vaccination?**		
No	Ref	
Yes	29.43 (3.34–259.41)	0.002
**Self-Efficacy** (ability to take care of one’s own health)		
No	Ref	
Yes	1.04 (0.84–1.29)	0.715
**Health Literacy**		
Low	Ref	
High	1.46 (0.74–2.90)	0.276
**Religion**		
None	Ref	
Catholic	0.98 (0.39–2.47)	0.965
Other	1.07 (0.13–9.41)	0.948
I prefer not to answer	0.25 (0.02–3.06)	0.278
**Political orientation**		
Moderate	Ref	
Left	0.51 (0.21–1.21)	0.124
Right	1.37 (0.55–3.42)	0.505
I prefer not to answer	2.29 (0.85–6.22)	0.103
**Perceived quality of the Healthcare System**		
Poor (low quality)	Ref	
Average (medium quality)	2.44 (0.81–7.41)	0.115
Good (high quality)	1.85 (0.61–5.56)	0.275
**Knowledge of HPV vaccination**		
Limited	Ref	
Good	1.56 (0.73–3.35)	0.250
**Attitudes towards HPV vaccination**		
Negative	Ref	
Positive	4.56 (1.65–12.60)	0.003

* outcome of interest: vaccinated children (*n* = 112, 47.9%) vs. unvaccinated children (*n* = 122, 52.1%). Ref: reference category.

**Table 7 vaccines-13-00599-t007:** Results of the binary logistic regression (Model 3). Factors associated with the intention to vaccinate children <12 yo *.

Variables	OR (95% CI)	*p*-Value
**Age**	1.01 (0.99–1.04)	0.232
**Gender**		
Male	Ref	
Female	0.67 (0.32–1.45)	0.312
**Educational level**		
High school or lower	Ref	
University degree	0.56 (0.23–1.37)	0.205
Postgraduate	0.78 (0.16–3.84)	0.759
**Ability to manage household financial resources**		
Definitely good/Fairly well	Ref	
With some difficulty/with many difficulties	1.03 (0.48–2.17)	0.948
**Occupation**		
Health workers	Ref	
Non-healthcare occupation	2.12 (0.36–12.42)	0.405
Housewives	1.67 (0.22–12.90)	0.619
Unemployed	1.95 (0.19–20.38)	0.577
**Source of information**		
No information	Ref	
Other HCWs	1.41 (0.50–3.95)	0.512
Pediatrician/gyn	5.86 (1.85–18.50)	0.003
Other	2.38 (0.78–7.25)	0.126
**Have you ever received HPV vaccination?**		
No	Ref	
Yes	4.07 (1.48–11.19)	0.006
**Self-Efficacy** (ability to take care of one’s own health)		
No	Ref	
Yes	0.94 (0.74–1.20)	0.609
**Health Literacy**		
Low	Ref	
High	1.53 (0.71–3.29)	0.103
**Religion**		
None	Ref	
Catholic	0.89 (0.31–2.53)	0.824
Other	1.53 (0.26–8.94)	0.639
I prefer not to answer	0.62 (0.08–4.73)	0.645
**Political orientation**		
Moderate	Ref	
Left	0.22 (0.07–0.66)	0.007
Right	0.69 (0.27–1.74)	0.426
I prefer not to answer	1.04 (0.36–2.96)	0.948
**Perceived quality of the Healthcare System**		
Poor (low quality)	Ref	
Average (medium quality)	4.09 (1.04–16.00)	0.043
Good (high quality)	2.54 (0.60–10.72)	0.293
**Knowledge of HPV vaccination**		
Limited	Ref	
Good	4.89 (2.21–10.84)	<0.001
**Attitudes towards HPV vaccination**		
Negative	Ref	
Positive	4.90 (2.02–11.84)	<0.001

* outcome of interest: high intention (*n* = 71, 26.4%) vs. low/moderate intention (*n* = 198, 73.6%). Ref: reference category

## Data Availability

The dataset generated and analyzed during the current study includes sensitive information on participants’ socio-demographic characteristics and personal attitudes and behaviors towards HPV vaccination. According to the informed consent obtained from participants, data were to be used exclusively in aggregated form for research purposes. Anonymized data may be made available by the corresponding author upon reasonable request and subject to approval by the appropriate ethics committee.

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
