# Peer review of "A Cross-Sectional Study to Understand HPV Vaccine Hesitancy and Influencing Factors in Italian Adults"

_vaccines, 2025, doi:10.3390/vaccines13060599_

Round 1

Reviewer 1 Report

Comments and Suggestions for Authors

The authors examined the factors influencing HPV vaccine acceptance involving adults aged 18 and older, with the data collected from a web-based survey using a structured questionnaire. I have the following issues for the authors to consider and improve the manuscript:  

Abstract

1. The objectives in line 12 are written too simplistically and don't align with the rest of the text. They should be strengthened to highlight the parent's viewpoint.

Introduction

  1. The section on male vaccination in line 40 needs to add specific age details.
  2. The transition between the second paragraph (low HPV vaccination rates among adolescents under the NIP) and the third paragraph (focus on adults’ KAP) feels abrupt. Consider clarifying the rationale for shifting the study population from adolescents to adults.

Materials and Methods

  1. In the "second section" describing variables (source of information, self-efficacy [SE], health literacy [HL]), specify whether these variables are HPV-specific (e.g., "sources of HPV vaccine information") or general (e.g., "general health information sources"). This distinction is critical for interpreting results.
  2. Align the order of variables in the Statistical analysis subsection with the Survey instrument sub This improves readability and logical flow.
  3. The use of p ≤0.25 as a threshold for variable inclusion in multivariable models is unconventional. Typically, stricter thresholds (e.g., p <0.05 or p <0.10) are applied to avoid overfitting. Justify this choice or cite methodological references supporting this approach.
  4. In Sample Size Calculation, the use of "perception of HPV vaccine safety" as a proxy for "vaccine hesitancy" requires justification. Why not directly use hesitancy/rejection rates? Explain how safety concerns alone sufficiently represent the broader construct of hesitancy.
  5. The criteria for Model 3 in line 122(age≥18) doesn’t align with the criteria in line 252(age>18).Please confirm the details and revise for consistency.
  6. Clarify the reasons behind omitting relevant occupations in lines131 and 133.
  7. Measurement is missing

Results

  1. There appears to be inconsistency: the mean age is reported as 54 years, yet the maximum age in the sample is also 54. This suggests either a data error. Verify and correct this.
  2. The inclusion of self-efficacy (SE) in Table 1 is puzzling since SE is typically an variable in section two along with the source of information and HL, not a demographic/professional characteristic.
  3. The three knowledge questions (prevents cervical cancer, prevents oral cancer, useful in males) focus narrowly on vaccine utility for specific cancers. This overlooks broader knowledge domains (e.g., transmission routes, recommended age, dosing schedule).
  4. The term ‘univariable analysis’ is the standard terminology in line 201.Please confirm the details and revise for consistency.
  5. Move Tables for the univariate analysis results to supplementary file
  6. Based on the questionnaire structure in Table 3, the analysis should employ a binary logistic regression model, not a multiple logistic regression model in line 238.
  7. Based on the questionnaire structure in Table 3, the analysis in line 250 should be conducted using multiple linear regression, not the current one. In summary, the title of section 3.4 should be revised to ‘Regression Analysis’.
  8. Section 3.4 should include offspring over 18 years old; however, according to the policy in 2007, this criterion must be limited to females under 30 years old.

8.To emphasize the parental perspective, the section should elaborate on parents' critical role in HPV vaccination procession, particularly regarding their influence on vaccination scheduling (e.g., ensuring timely doses).

Discussion

  1. To align with the entire Discussion section and enhance readability, a synthesizing statement should be inserted prior to lines 291-292.

Author Response

Comments and Suggestions for Authors

The authors examined the factors influencing HPV vaccine acceptance involving adults aged 18 and older, with the data collected from a web-based survey using a structured questionnaire. I have the following issues for the authors to consider and improve the manuscript

Response

Dear Reviewer,

We sincerely thank you for your thoughtful and constructive comments, which have greatly helped us improve our manuscript.

We have carefully considered all your remarks and revised the manuscript accordingly. Below, we provide a point-by-point response to each of your comments, detailing the changes made and clarifying any issues raised.

We hope that the revised version addresses your concerns and meets the journal’s standards.

Sincerely,

Marianna Riccio, Azzurra Massimi, Erika Renzi, Michele Innocenzio, Roberta Siliquini, Fabrizio Bert, Anna Odone, Carolina Marzuillo, Paolo Villari, Corrado De Vito 

Abstract

1.The objectives in line 12 are written too simplistically and don't align with the rest of the text. They should be strengthened to highlight the parent's viewpoint.

Response:
We appreciate the reviewer’s observation. We have revised the objective in the Abstract to reflect the parental perspective by emphasizing their role, the statement now appears as follows: “This study analyzed the factors influencing HPV vaccine acceptance among adults, including a focused analysis of parental behaviors” (lines 14,15).

Introduction

  1. The section on male vaccination in line 40 needs to add specific age details.

Response:
We thank the Reviewer for this comment. In the Introduction Section, we have clarified the target age group for the male HPV vaccination program. Specifically, we added that in 2017, the program was extended to include 12-year-old boys, starting with those born in 2006, aligning with the existing strategy for girls. (See manuscript, lines 47-50).

  1. The transition between the second and third paragraphs feels abrupt. Consider clarifying the rationale for shifting the study population from adolescents to adults.

Response: 
Thanks for the suggestion. We first address adolescent vaccination coverage, the primary target group, followed by a broader discussion of vaccine hesitancy, as it represents a major barrier to achieving adequate coverage both in the general population and specific subgroups, such as parents. However, transitional text has been added to clarify this rationale in the revised manuscript.  (See manuscript, lines 55-62).

Materials and Methods

  1. In the "second section" describing variables (source of information, self-efficacy [SE], health literacy [HL]), specify whether these variables are HPV-specific (e.g., "sources of HPV vaccine information") or general (e.g., "general health information sources"). This distinction is critical for interpreting results.

Response:
Thank you for pointing this out. The information source variable refers exclusively to HPV-related content; we clarified this in the Methods section: “sources of information about HPV vaccination” (lines 87-88 of the Survey Instrument subsection).
In contrast, the descriptions of self-efficacy (SE) and health literacy (HL) in the same section clearly indicate that these were assessed in a general context and are not specific to HPV (see lines 89-91).

  1. Align the order of variables in the Statistical analysis subsection with the Survey instrument sub This improves readability and logical flow.

Response:
We appreciate this suggestion. We have revised the order of variables as indicated, thereby enhancing readability and logical flow. Specifically, we moved the description of the information source variable to follow the socio-demographic characteristics within the Statistical Analysis section (see lines 120-123). The order of the remaining variables is consistent with the Survey instrument section.

  1. The use of p ≤0.25 as a threshold for variable inclusion in multivariable models is unconventional. Typically, stricter thresholds (e.g., p <0.05 or p <0.10) are applied to avoid overfitting. Justify this choice or cite methodological references supporting this approach.

Response:      
We thank the Reviewer for this insightful comment. Including variables with a p-value ≤0.25 in the multivariable models was based on the strategy suggested by Hosmer and Lemeshow, who recommend this threshold to avoid excluding potentially important variables at an early stage of model building.

To address this, we have added the following clarification in the Methods section (lines 150-151): “Three regression models were developed based on the strategy suggested by Hosmer and Lemeshow”. [Hosmer DW, Lemeshow S: Applied Logistic Regression. 2000, New York: Wiley & Sons.]

We acknowledge the importance of model overfitting. According to the strategy mentioned above, variables were included in the model not only based on statistical criteria but also considering epidemiological relevance and findings from previous literature. All selected variables were retained in the final model regardless of their statistical significance, as the analysis objective was explanatory rather than predictive. This strategy aims to ensure proper control for potential confounding and to provide a more accurate interpretation of the associations under study.

Therefore, we have already added a statement in the Methods section (see manuscript, lines 153-157) specifying that multicollinearity and model fit were assessed to reduce the risk of overfitting.

  1. In Sample Size Calculation, the use of "perception of HPV vaccine safety" as a proxy for "vaccine hesitancy" requires justification. Why not directly use hesitancy/rejection rates? Explain how safety concerns alone sufficiently represent the broader construct of hesitancy.

Response:
We acknowledge the limitation of not directly measuring vaccine hesitancy, but it is a complex and multifaceted construct typically assessed through various methodologies. However, our analysis of all samples primarily focused on attitudes. For this reason, we selected a specific attitude -trust in vaccine safety- which was assessed across all sample. This item is widely used in previous research and defines the 'confidence' dimension of the well-established '3 Cs' model. Additionally, we have revised the sample size section to improve clarity. (see manuscript, lines 175-179)

  1. The criteria for Model 3 in line 122 (age≥18) doesn’t align with the criteria in line 252(age>18). Please confirm the details and revise for consistency.

Response:
Thanks for this observation. The age criteria have been revised throughout the manuscript where appropriate. Specifically, we have corrected the classifications as follows: children aged 12–17 and children aged 18 and above. Regarding Model 3, we included parents of children aged < 12.

  1. Clarify the reasons behind omitting relevant occupations in lines 131 and 133.

Response:
Thank you for the suggestion. We have revised the manuscript to include a more detailed explanation of why certain categories were omitted from the regression models. A new statement has now been added in lines 163-169: “In Model 1, the gender category "I prefer not to answer" included only three observations. In Model 2, the gender category "I prefer not to answer" and the occupational category "students" comprised a single observation each. In Model 3, the occupational categories "students" and "retired" each included two observations. Due to the extremely limited sample sizes, these categories were excluded from the regression analyses to avoid statistical and methodological issues.”

  1. Measurement is missing.

Response:
We acknowledge the Reviewer’s concern regarding the lack of a description of the measurement. In response, we reorganized the Methods section to describe all measurement tools used for each construct explicitly.

Results

  1.  There appears to be inconsistency: the mean age is reported as 54 years, yet the maximum age in the sample is also 54. This suggests a data error. Verify and correct this.

Response:
Thank you for noticing this. We have reviewed and corrected the reported descriptive statistics. The discrepancy was due to an error in the Results section. The corrected values are now reported accurately in Table 1.

  1. The inclusion of self-efficacy (SE) in Table 1 is puzzling since SE is typically a variable in section two along with the source of information and HL, not a demographic/professional characteristic.

Response:
Thanks for the comment. We have now moved SE, HL and source of information to a new specific table (Table 2), separate from the demographic/professional characteristics table, as suggested. Therefore, we have also removed the Fig.1 and STable 2 about the description of information sources.

  1. The three knowledge questions (prevents cervical cancer, prevents oral cancer, useful in males) focus narrowly on vaccine utility for specific cancers. This overlooks broader knowledge domains (e.g., transmission routes, recommended age, dosing schedule).

Response:
We appreciate this insightful comment. We chose to focus the knowledge assessment on cancer-related outcomes because one of the most critical aspects of HPV vaccination is its role in cancer prevention. This association is often under-recognized by the general population and not intuitively linked to infectious diseases. This focus also allowed us to keep the questionnaire concise and reduce the number of questions.

  1. The term ‘univariable analysis’ is the standard terminology in line 201. Please confirm the details and revise for consistency.

Response:
We thank the reviewer for the terminology suggestion. To be consistent with standard statistical language, we have replaced the term with “univariable analysis.” (see lines 17, 245,246)

  1. Move Tables for the univariate analysis results to supplementary file.

Response:
Thank you for the suggestion. We have now moved the univariable analysis table (Table 4) to the Supplementary Material (now STable 5) and referenced it appropriately in the main text. (see line 246)

  1. Based on the questionnaire structure in Table 3, the analysis should employ a binary logistic regression model, not a multiple logistic regression model in line 238.

Response:
We have revised the text to specify that binary and multivariable logistic regression was used, as appropriate for the dichotomous outcome variable and the inclusion of multiple independent variables (see lines 271, 388, Table 5,6,7).

  1. Based on the questionnaire structure Table 3, the analysis in line 250 should be conducted using multiple linear regression, not the current one. In summary, the title of section 34 should be revised to ‘Regression Analysis’.

Response:
We thank the reviewer for this comment. We conducted a multivariable logistic regression analysis in all three models. For Model 3, the outcome variable—intention to vaccinate children (under 12) in the future, originally measured on a 0–100 scale—was dichotomized to allow for logistic regression. This was described in section 2.3.1. (see lines 131-133), and now also included in STable 1.
To enhance clarity, we have revised the notes of the regression model tables to provide a clearer explanation of the outcome definitions.
Note of Table 5: “outcome of interest: positive attitudes (343, 18.8%) vs negative attitudes (1.478, 81.2%)”; note of Table 6: “outcome of interest: vaccinated children (n=112, 47.9%) vs unvaccinated children (n=122, 52.1%)”; note of Table 7: “outcome of interest: high intention (n=71, 26.4%) vs low/moderate intention (n=198, 73.6%).”

This approach was chosen to facilitate logistic regression analysis and is consistent with similar studies on intention-to-vaccinate outcomes

  1. Section 3.4 should include offspring over 18 years old; however, according to the policy in 2007, this criterion must be limited to females under 30 years old.

Response:
We thank the Reviewer for this comment. In our analysis, we focused on the age group currently most relevant to HPV vaccination policies—adolescent girls and boys. More detailed stratification was not feasible, as the dataset provided only categorical age groups (<12 years, 12–18 years, and >18 years) and did not include the quantitative variable. Furthermore, data on the sex of the children were not available. The only indirect indication of child sex appeared in some participants’ responses regarding reasons for non-vaccination, where having a son was mentioned as a justification.

  1. To emphasize the parental perspective, the section should elaborate on parents' critical role in HPV vaccination procession, particularly regarding their influence on vaccination scheduling (e.g., ensuring timely doses).

Response:
We agree that the parental role is important. In the Discussion section, we highlighted the role of parents in HPV vaccination uptake through decision-making, which aligns with the study’s objectives. Therefore, the following statement was added:
 “Additionally, the subgroup analysis allowed for a comprehensive evaluation of the specific characteristics influencing parents’ practices, underscoring their pivotal role in both initiating and adhering to HPV vaccination schedules. As primary health decision-makers for their children, they significantly affect vaccine acceptance and uptake.” (lines 326-331).
“….among adolescents - the primary target group—and their parents, who are responsible for vaccination decisions” (lines 395-397).

Discussion

   1. To align with the entire Discussion section and enhance readability, a synthesizing statement should be inserted prior to lines 291-292.

Response:
Thank you for this constructive suggestion. A synthesizing statement has been added at the outset of this section to guide the forthcoming discussion: “Our findings highlight HL's essential role in shaping attitudes towards HPV.” (line 349)

Reviewer 2 Report

Comments and Suggestions for Authors

In this manuscript authors have discussed about HPV vaccine use and hesitancy in Italian population. It can be helpful for a country like Italy and will increase awareness among people. But the population and study time is very small, lack of inclusion and exclusion criteria, medical history and status of parents (were they aware or not) etc. Instead of table they should have shown some figures.

Author Response

Comments and Suggestions for Authors

In this manuscript authors have discussed about HPV vaccine use and hesitancy in Italian population. It can be helpful for a country like Italy and will increase awareness among people. But the population and study time is very small, lack of inclusion and exclusion criteria, medical history and status of parents (were they aware or not) etc. Instead of table they should have shown some figures.

Response

Dear Reviewer,

We sincerely thank you for your constructive feedback and for recognizing the relevance of our manuscript.
Regarding the sample size, we would like to clarify that our study involved a relatively large and demographically diverse sample (n = 1,821), which exceeds the minimum sample size calculated a priori and enhances the generalisability of our findings.
As for the study population, participants were adults aged 18 years or older, residing in Italy, and capable to completing an online questionnaire. These inclusion criteria were chosen to reflect the general adult population and are explicitly stated in the revised Methods section.
We believe that timing-related issues are not applicable, as the study has a cross-sectional design.
We acknowledge the Reviewer’s suggestion regarding additional variables, such as participants’ medical history and parental awareness of HPV vaccination. We agree that including such data could have enriched the analysis. However, due to the study’s design and objectives, and to maintain respondent burden at a manageable level, these variables were not collected.
Finally, with respect to the presentation of results, we opted for tables rather than figures to ensure clarity and allow for the detailed reporting of variables and statistical findings. Nevertheless, we remain open to including figures should the editorial team consider it beneficial for readers.

Once again, we thank the Reviewer for their valuable comments, which have helped us improve the clarity and robustness of our manuscript.

Sincerely,

Marianna Riccio, Azzurra Massimi, Erika Renzi, Michele Innocenzio, Roberta Siliquini, Fabrizio Bert, Anna Odone, Carolina Marzuillo, Paolo Villari, Corrado De Vito

Reviewer 3 Report

Comments and Suggestions for Authors

Major Issues

  1. Psychometric Properties of the Instrument

    • The manuscript does not report any validation statistics for the survey instrument beyond referencing prior studies. Please clarify whether internal consistency (e.g., Cronbach’s alpha) or construct validity was assessed for scales measuring knowledge, attitudes, or self-efficacy in your sample.

  2. Inclusion/Exclusion Criteria

    • There is no mention of inclusion or exclusion criteria for participation in the survey. This information is necessary to assess the sample's representativeness and potential selection bias.

  3. Recruitment and Sampling Strategy

    • The manuscript does not adequately explain how participants were selected from the Norstat panel. Was any stratified or quota sampling used? Were respondents incentivised? Please clarify.

  4. Response Rate

    • The response rate is not reported. This is essential for evaluating non-response bias and should be included in the Methods section.

  5. Inappropriate Structuring of Methods Section

    • The Statistical Analysis section (Lines 93–115) includes details about variable definitions and measurement categorisations, which would be better placed in a dedicated Variables and Measures subsection. Please revise accordingly and follow the STROBE guidelines strictly.

  6. Sample Size Concerns

    • While exceeding the required sample size (718 vs. 1821) can enhance power, it also raises the risk of identifying statistically significant associations that are not practically meaningful (i.e., increased risk of Type I error). This should be acknowledged in the Discussion.

  7. Sample Size Calculation

    • The explanation of the sample size calculation (Lines 144–153) is overly embedded within the analysis section. It should be moved into its own clearly labelled subsection within Methods.

  8. Variable Selection for Multivariable Models

    • Line 123: The decision to include variables with p < 0.25 from univariate analysis in multivariable models should be justified with appropriate references or methodological reasoning. It may also be worth noting whether model overfitting was assessed.

  9. Occupation Variable Classification

    • In Table 1, the occupational categories are unconventional (e.g., housewives, students, retirees) and not mutually exclusive with broader classifications such as healthcare vs. non-healthcare roles. Please justify this categorisation or consider reclassifying based on standard occupational groupings.

  10. Zero-Frequency Categories

    • The inclusion of professions such as “pharmacist” and “obstetrician” in Table 1 despite having zero observations is not informative and unnecessarily clutters the table. These should be removed or footnoted accordingly.

  11. Statistical Testing Approach

    • In Tables 3 and 4, it is unclear why both t-tests and Mann–Whitney U tests were used simultaneously. These tests serve different purposes depending on the distribution of the data. Please clarify whether normality was assessed and which test was ultimately applied for each variable.

  12. Causal Language in Cross-Sectional Study

    • Some statements in the abstract and discussion imply causation (e.g., “vaccination increased the likelihood…”). Given the cross-sectional design, such claims should be carefully reworded to reflect associative—not causal—interpretations.

  13. Regression Modelling Structure

    • The rationale for running three separate models (general population, parents of 12–18, parents of <12) is sound. However, the interrelation between knowledge, attitudes, intention, and behaviour is complex. A conceptual framework (e.g. WHO’s Vaccine Hesitancy Matrix, the 5C model, or the Health Belief Model) would strengthen the interpretation and structure of your models.

Minor Issues

  1. Abstract Results Incomplete

    • While the abstract notes that logistic regression was used (Lines 15–16), the results section (Lines 17–23) does not report any numerical findings. Please include key effect estimates (e.g., odds ratios with confidence intervals) to substantiate claims.

  2. Grammatical Correction (Line 30)

    • The sentence “It poses a significant public health issue due…” should be revised to: “It poses a significant public health concern due to its established association with various cancers.”

  3. p-Value Notation

    • Throughout the manuscript, “pV” is used to denote p-values. This is non-standard. Please replace all instances with the standard abbreviation “p”.

  4. Tables 5, 6, and 7 – Language Editing

    • Several rows in these tables contain inconsistent or awkward phrasing (e.g., “Ref”, "Continue", inconsistent use of spacing). Ensure all tables follow journal formatting guidelines and are reviewed for grammar and clarity.

  5. Terminology Clarification

    • The manuscript occasionally uses “univariate univariable analysis,” which is redundant. The correct term is “univariable” or “univariate” depending on the context.

  6. Health Literacy Measurement Tool

    • The SILS is a simple screening tool and does not measure comprehensive health literacy domains. Please mention this limitation in the Discussion.

  7. Sociopolitical Determinants

    • Findings regarding political orientation and religion are intriguing but underexplored. A brief discussion on how these may interact with trust in health systems and vaccination attitudes would add depth.

Author Response

Response

Dear Reviewer,

We sincerely thank you for your comprehensive and insightful comments, which have contributed substantially to improving the clarity, structure, and scientific rigor of our manuscript.

Please find below our point-by-point responses, along with the corresponding revisions made to the manuscript. We have also revised the language for clarity and fluency, employing AI-assisted tools (ChatGPT) to enhance the English language and overall readability of the text.

We hope that the revised version addresses your concerns satisfactorily and meets the journal’s standards.

Sincerely,

Marianna Riccio, Azzurra Massimi, Erika Renzi, Michele Innocenzio, Roberta Siliquini, Fabrizio Bert, Anna Odone, Carolina Marzuillo, Paolo Villari, Corrado De Vito

Comments and Suggestions for Authors

Major Issues

  1. Psychometric Properties of the Instrument
    The manuscript does not report any validation statistics for the survey instrument beyond referencing prior studies. Please clarify whether internal consistency (e.g., Cronbach’s alpha) or construct validity was assessed for scales measuring knowledge, attitudes, or self-efficacy in your sample.

Response:
We appreciate this valuable observation. In response, we have reported the internal consistency of the attitude and knowledge scales using Cronbach’s alpha within our sample. The results showed a Cronbach’s alpha of 0.82 for the knowledge scale (3 items) and 0.61 for the attitudes scale (4 items). While the alpha for attitudes is slightly below the conventional threshold of 0.70, it is still considered acceptable in applied research, especially given the small number of items and the conceptual complexity of attitudinal constructs. Previous literature [Nunnally JC, Bernstein IH: Psychometric Theory (3rd ed.). 1994, New York: McGraw-Hill.] supports the use of alpha values above 0.60 in such contexts. Furthermore, item-level analysis indicated no single item whose removal would substantially improve the scale’s reliability. Therefore, we retained all items and considered the internal consistency acceptable for the purposes of this study.
We have added the specific statement about Cronbach’s alpha in lines 99-105, with a new reference [29]. Conversely, the self-efficacy assessment was based on an instrument largely used on different populations.

  1. Inclusion/Exclusion Criteria
    There is no mention of inclusion or exclusion criteria for participation in the survey. This information is necessary to assess the sample's representativeness and potential selection bias

Response:
Thank you for the comment. As for the study population, participants were adults aged 18 years or older, residing in Italy, and capable to completing an online questionnaire. These inclusion criteria were chosen to reflect the general adult population and are explicitly stated in the revised subsection 2.1.

  1. Recruitment and Sampling Strategy
    The manuscript does not adequately explain how participants were selected from the Norstat panel. Was any stratified or quota sampling used? Were respondents incentivised? Please clarify.

Response:
We appreciate this insightful comment. We have reviewed subsection 2.1 (lines 77-79).
A professional panel provider (Norstat-Italia srl) was hired to recruit, through a stratified sampling strategy (by age, gender, and geographical area of residence in Italy) a random sample from all Italian regions. To be included in the study, participants had to live in Italy and be at least 18 years old. Participation was voluntary, and respondents received standard panel incentives as per Norstat's policies.
For more detailed information, please refer to the Norstat Panel website: https://www.norstatpanel.com/it.

  1. Response Rate
    The response rate is not reported. This is essential for evaluating non-response bias and should be included in the Methods

Response:
We thank the Reviewer for this helpful comment. As suggested, we have clarified that the survey was originally designed to address multiple research objectives. Consequently, a larger sample size was targeted to enable subgroup analyses and support other ongoing studies. This accounts for the relatively high number of participants included in the present analysis and justifies the sampling strategy adopted. The following statement was added in the Methods section, lines 187-190: “The survey was initially designed to address multiple research objectives beyond the scope of the present analysis. Therefore, a larger sample size was targeted to ensure sufficient statistical power across different planned investigations and to allow for subgroup analyses.”

  1. Inappropriate Structuring of Methods Section
    The Statistical Analysissection (Lines 93–115) includes details about variable definitions and measurement categorisations, which would be better placed in a dedicated Variables and Measures Please revise accordingly and follow the STROBE guidelines strictly.

Response:
Thank you for the suggestion. We have restructured the Methods section to include a dedicated “Variables and Measures” subsection (line 110), where we describe the operational definitions, coding, and categorisation of all key variables. The “Statistical Analysis” section now presents new reorganisation to improve clarity and ensures compliance with STROBE recommendations.

  1. Sample Size Concerns
    While exceeding the required sample size (718 vs. 1821) can enhance power, it also raises the risk of identifying statistically significant associations that are not practically meaningful (i.e., increased risk of Type I error). This should be acknowledged in the Discussion.

Response:
We thank the Reviewer for this insightful observation. We acknowledge that exceeding the required sample size may increase the risk of detecting statistically significant associations that are not necessarily of practical relevance, thereby potentially inflating the risk of Type I error. In response, we have added the following sentence to the Discussion section to acknowledge this limitation and to emphasize the need for cautious interpretation of findings with borderline significance (lines 423-426): “Additionally, while the large sample size increased the statistical power of our analyses, it may also heighten the chance of detecting statistically significant associations that lack substantial practical importance. Therefore, interpreting results, especially those with borderline significance, should be cautiously approached.”
Finally, to enhance clarity, we have included the total number of participants in the abstract (line 18)

  1. Sample Size Calculation
    The explanation of the sample size calculation (Lines 144–153) is overly embedded within the analysis section. It should be moved into its own clearly labelled subsection within Methods.

Response:
We have now moved the sample size calculation to a new, clearly labelled subsection titled “Study Size” (line 170) under the Statistical Analysis section.

  1. Variable Selection for Multivariable Models
    Line 123: The decision to include variables with p < 0.25 from univariate analysis in multivariable models should be justified with appropriate references or methodological reasoning. It may also be worth noting whether model overfitting was assessed.

Response:
We thank the Reviewer for this insightful comment. Including variables with a p-value ≤0.25 in the multivariable models was based on the strategy suggested by Hosmer and Lemeshow, who recommend this threshold to avoid excluding potentially important variables at an early stage of model building. To address this, we have added the following clarification in the Methods section (lines 152-153, with a new reference [31]): “Three regression models were developed based on the strategy suggested by Hosmer and Lemeshow”. [Hosmer DW, Lemeshow S: Applied Logistic Regression. 2000, New York: Wiley & Sons.]

We acknowledge the importance of model overfitting. According to the strategy mentioned above, variables were included in the model not only based on statistical criteria but also considering epidemiological relevance and findings from previous literature. All selected variables were retained in the final model regardless of their statistical significance, as the analysis objective was explanatory rather than predictive. This strategy aims to ensure proper control for potential confounding and to provide a more accurate interpretation of the associations under study.
Therefore, we have already added a statement in the Methods section (see manuscript, lines 153-157) specifying that multicollinearity and model fit were assessed to reduce the risk of overfitting.

  1. Occupation Variable Classification
    In Table 1, the occupational categories are unconventional (e.g., housewives, students, retirees) and not mutually exclusive with broader classifications such as healthcare vs. non-healthcare roles. Please justify this categorisation or consider reclassifying based on standard occupational groupings.

Response:
We thank the Reviewer for highlighting this point. However, in our categorisation, we considered these occupational groups to be mutually exclusive. For instance, individuals identified as housewives were not simultaneously employed or classified as healthcare workers. Similarly, retired individuals were not grouped with the unemployed or with those identified as homemakers.

  1. Zero-Frequency Categories
    The inclusion of professions such as “pharmacist” and “obstetrician” in Table 1 despite having zero observations is not informative and unnecessarily clutters the table. These should be removed or footnoted accordingly.

Response:
Thank you for pointing this out. We have removed the zero-frequency categories from Table 1 to improve clarity.

  1. Statistical Testing Approach
    In Tables 3 and 4, it is unclear why both t-tests and Mann–Whitney U tests were used simultaneously. These tests serve different purposes depending on the distribution of the data. Please clarify whether normality was assessed and which test was ultimately applied for each variable.

Response:
Normality was assessed prior to analysis using the Shapiro-Wilk test. Given the sample size and the statistical power of the parametric test, the results were evaluated using the independent samples t-test and confirmed by the Mann-Whitney U test in cases of deviation from normality.

  1. Causal Language in Cross-Sectional Study
    Some statements in the abstract and discussion imply causation (e.g., “vaccination increased the likelihood…”). Given the cross-sectional design, such claims should be carefully reworded to reflect associative—not causal—interpretations.

Response:
Thank you for highlighting this concern. We have carefully reviewed and edited the manuscript to remove or rephrase any causal language. (lines 22-25, 267-269, 294-295, 307-309, 313-315, 385-397)

  1. Regression Modelling Structure
    The rationale for running three separate models (general population, parents of 12–18, parents of <12) is sound. However, the interrelation between knowledge, attitudes, intention, and behaviour is complex. A conceptual framework (e.g. WHO’s Vaccine Hesitancy Matrix, the 5C model, or the Health Belief Model) would strengthen the interpretation and structure of your models.

Response:
We thank the Reviewer for the interesting suggestion. Given the structure of our questionnaire and the variables analyzed (trust in health workers, levels of knowledge, perceived risk, sources of information), we considered the 5C model to be the most appropriate and consistent with our data. Although not all components of the model were directly measured, the inclusion of this framework allows for a clearer understanding of the factors underlying vaccine hesitancy in the study population. We briefly describe the model in the Methods section (see lines 135-141, with a new reference [30]) and cite it in the Discussion (lines 408-415) to support the interpretation of our results.

Minor Issues

  1. Abstract Results Incomplete
    While the abstract notes that logistic regression was used (Lines 15–16), the results section (Lines 17–23) does not report any numerical findings. Please include key effect estimates (e.g., odds ratios with confidence intervals) to substantiate claims.

Response:
We appreciate this insightful comment. We have revised the abstract to include key numerical findings from the logistic regression analyses, including ORs and 95% CIs for significant predictors.

  1. Grammatical Correction (Line 30)
    The sentence “It poses a significant public health issue due…” should be revised to: “It poses a significant public health concern due to its established association with various cancers.”

Response:
The sentence has been revised as suggested (lines 36-37).

  1. p-Value Notation
    Throughout the manuscript, “pV” is used to denote p-values. This is non-standard. Please replace all instances with the standard abbreviation “p”.

Response:    
We have carefully reviewed the abbreviation “p”.

  1. Tables 5, 6, and 7 – Language Editing
    Several rows in these tables contain inconsistent or awkward phrasing (e.g., “Ref”, "Continue", inconsistent use of spacing). Ensure all tables follow journal formatting guidelines and are reviewed for grammar and clarity.

Response:
The language and formatting of Tables 5, 6, and 7 have been reviewed and corrected to ensure consistency: we removed ‘continue’ and we added a note at the end of each table to clarify that 'Ref' denotes the reference category.

  1. Terminology Clarification
    The manuscript occasionally uses “univariate univariable analysis,” which is redundant. The correct term is “univariable” or “univariate” depending on the context.

Response
We thank the Reviewer for the terminology suggestion. We have replaced redundant or incorrect uses of “univariate univariable analysis” with the appropriate term “univariable” (see lines 17, 245,246)

  1. Health Literacy Measurement Tool
    he SILS is a simple screening tool and does not measure comprehensive health literacy domains. Please mention this limitation in the Discussion.

Response:
We agree that the SILS is a brief screening tool rather than a comprehensive measure. However, we would like to emphasize that, firstly, health literacy (HL) was not the primary focus of our study, but rather one of the determinants under investigation. Secondly, existing literature supports the use of the SILS as a valid alternative to more complex instruments for assessing HL. Nevertheless, we have acknowledged in the limitations section (see manuscript, lines 429-431) that HL can be assessed using more comprehensive tools, which may allow for more robust conclusions.

  1. Sociopolitical Determinants
    Findings regarding political orientation and religion are intriguing but underexplored. A brief discussion on how these may interact with trust in health systems and vaccination attitudes would add depth.

Response:
Thank you for the opportunity to further elaborate on this point. These factors were not explored in greater detail in our analysis, as the focus was placed on other determinants that demonstrated stronger associations. However, several studies (as cited in our references) didn’t show these factors as significant predictors.
Nevertheless, we acknowledge their relevance as important contextual elements within the vaccine hesitancy (VH) framework. To address this, we have added a dedicated discussion in the revised manuscript to provide further insight and completeness (see manuscript, lines 402-407).

Round 2

Reviewer 3 Report

Comments and Suggestions for Authors

Dear author(s),

Thank you for your esteemed efforts in addressing my prior comments and amending the manuscript accordingly. I believe that the manuscript is in good shape now for publishing.

Sincerely,